

# Bulk Renormalization Group Flows
# and Boundary States in Conformal Field Theories

## John Cardy[1,2]⋆

**1** Department of Physics, University of California, Berkeley CA 94720, USA
**2** All Souls College, Oxford OX1 4AL, UK

⋆ cardy@berkeley.edu

## Abstract

We propose using smeared boundary states $e^{-\tau H}|\mathscr{B}\rangle$ as variational approximations to the ground state of a conformal field theory deformed by relevant bulk operators. This is motivated by recent studies of quantum quenches in CFTs and of the entanglement spectrum in massive theories. It gives a simple criterion for choosing which boundary state should correspond to which combination of bulk operators, and leads to a rudimentary phase diagram of the theory in the vicinity of the RG fixed point corresponding to the CFT, as well as rigorous upper bounds on the universal amplitude of the free energy. In the case of the 2d minimal models explicit formulae are available. As a side result we show that the matrix elements of bulk operators between smeared Ishibashi states are simply given by the fusion rules of the CFT.

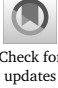
# 1 Introduction

Conformal field theories (CFTs) are supposed to correspond to the non-trivial renormalization group (RG) fixed points of relativistic quantum field theories (QFTs). Such theories typically contain a number of scaling operators of dimension $\Delta < d$ (where $d$ is the space-time dimension), which, if added to the action, are relevant and drive the theory to what is, generically, a trivial fixed point. The points along this trajectory then correspond to a massive QFT. In general there is a multiplicity of such basins of attraction of the RG flows, but enumerating them and determining which combinations of relevant operators lead to which basins, and therefore to what kind of massive QFT, in general requires non-perturbative methods. This problem is equivalent to mapping out the phase diagram in the vicinity of the critical point corresponding to the CFT.

Another way of characterizing these massive theories is through the analysis of the possible boundary states of the CFT. Imagine the scenario in which the relevant operators are switched on in only a half-space, say $x_0 < 0$. This will then appear as some boundary condition on the CFT in $x_0 > 0$. However the boundary conditions themselves undergo RG flows, with fixed points corresponding to so-called conformal boundary conditions. Therefore on scales $\sim M^{-1}$, where $M$ is the mass scale of the perturbed theory, the correlations near the boundary should be those of a conformal boundary condition, deformed by *irrelevant* boundary operators.

A similar question is raised through recent work on the spectrum of the entanglement hamiltonian in massive QFTs [1]. If the theory is defined in $\mathbf{R}^D$ and is in its ground state, and we study the entanglement between the degrees of freedom in the half-space $A : x_1 > 0$ and its complement, then the entanglement, or modular, hamiltonian $K_A = -(1/2\pi)\log\rho_A$, where $\rho_A$ is the reduced density matrix of $A$, takes the form [2,3]

$$K_A = \int_{x_1>0} x_1 T_{00}(x) d^D x \,, \tag{1}$$

which is nothing but the generator of rotations in euclidean space, or of boosts in lorentzian signature. In 1+1 dimensions we may consider a conformal transformation $z = x_1 + i x_0 = \epsilon e^w$ which sends the euclidean $z$-plane, punctured at the origin by a disc of radius $\epsilon$ representing the UV cutoff, to an semi-infinite cylinder of circumference $2\pi$. $K_A$ is then simply the generator of translations around this cylinder. However, if the QFT corresponds to a perturbed CFT, it is not conformally invariant, but rather the couplings transform as

$$\lambda \to \lambda \epsilon^{2-\Delta} e^{(2-\Delta)\mathrm{Re}\,w} \,, \tag{2}$$

where $\Delta < 2$ is the scaling dimension of the perturbing operator. Thus the dimensionless coupling $g = \lambda\epsilon^{2-\Delta}$ is effectively switched on over a length scale $O(1)$ near $\mathrm{Re}\,w \sim \log(1/g)$. If we are interested the low-lying spectrum of $K_A$, corresponding to the Rényi entropies $\mathrm{Tr}\,\rho_A^n$ with $n \gg 1$, the effective circumference of the cylinder is $2\pi n$ and we are then in a similar situation to the above, where the massive theory for $\mathrm{Re}\,w > \log(1/g)$ acts as an effective boundary condition on the CFT in $\mathrm{Re}\,w < \log(1/g)$. As concluded in [1], the low-lying spectrum of $K_A$ should therefore be that of the (boundary) CFT, with an appropriate boundary condition depending on the bulk perturbation.

The same question arises in the context of quantum quenches [4]. In this case we are interested in the real time evolution of an initial state $|\Psi_0\rangle$ under a hamiltonian $H$ of which it is not an eigenstate. An example is the case where $H = H_{CFT}$ and $|\Psi_0\rangle$ is the ground state of the massive perturbed CFT. This is a difficult problem, and in [4,5] the step was taken of replacing this ground state by a conformal boundary state perturbed by irrelevant operators.[1]

---

[1]In [4] only the smeared states of the form (4) were considered, which happen to lead to subsystem thermalization, while in [5] it was argued that more general states should lead to a generalized Gibbs ensemble.

This allows the explicit computation of the imaginary time evolution and the continuation to real time, which would be very difficult for the exact ground state of the massive theory.

Thus an important problem in all these cases is to determine to which conformal boundary condition a particular combination of bulk operators should correspond. For simple examples this is apparent by physical inspection. For example, the CFT corresponding to the critical Ising model has two relevant operators, coupling to the magnetic field $h$ and the deviation $t$ of the reduced temperature from its critical value. There are three stable RG fixed points at $(h = 0, t \to +\infty)$ and $h \to \pm\infty$, respectively the sinks for the disordered and the two ordered phases, and corresponding to the three conformal boundary conditions when the Ising spins are respectively free and fixed, either up or down.

One way to make this identification is to think of the boundary condition as defining a state $|\mathscr{B}\rangle$ when the theory is quantized on a time slice $x_0 = $ constant. In that language we may regard the perturbed CFT as described by a hamiltonian operator

$$\hat{H} = \hat{H}_{CFT} + \sum_j \lambda_j \int \hat{\Phi}_j(x) d^{d-1} x \,, \tag{3}$$

where the $\{\hat{\Phi}_j\}$ are relevant operators. We then ask which $|\mathscr{B}\rangle$, suitably deformed by boundary irrelevant operators, is closest in some sense to the ground state of $\hat{H}$ at strong coupling.

Conformal boundary states by themselves contain no scale, and therefore cannot be good candidates for the ground state of $\hat{H}$. Indeed, they must have infinite energy compared to this state. In known examples in 2d (and, for example, for free theories in higher dimensions) they are also non-normalizable. We must therefore deform them by irrelevant boundary operators in order to give them a scale. The simplest such operator is the stress tensor $\hat{T}_{00}$, which has scaling dimension $d$ and therefore boundary RG eigenvalue $(d-1) - d = -1$. Since its space integral is the CFT hamiltonian, including only this operator is tantamount to considering boundary states 'smeared' by evolution in imaginary time:

$$e^{-\tau \hat{H}_{CFT}} |\mathscr{B}\rangle \,, \tag{4}$$

where $\tau > 0$ is parameter with the dimensions of length. Such states have finite energy and correlation length $\propto \tau^{-1}$, and also finite norm.

Such smeared boundary states may be thought of as a continuum version of matrix product states (MPS). Indeed, a lattice discretization of the euclidean path integral, illustrated in Fig. 1, suggests that such states correspond to matrices with internal dimension $\sim N^{\tau/\delta\tau}$, where $N$ is the number of states on each lattice edge and $\delta\tau$ is the time step. However, unlike discrete MPS states, the smeared boundary state (4) automatically has the correct short-distance behavior of the CFT.

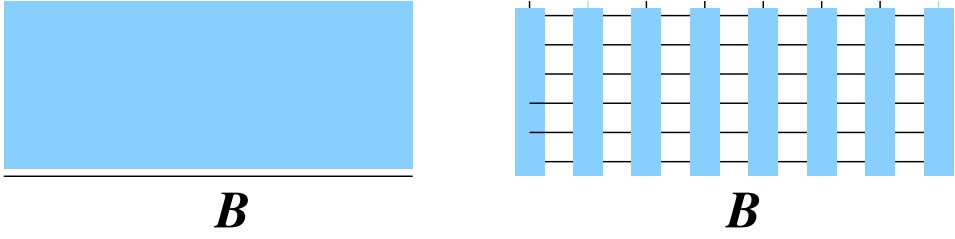

Figure 1: Path integral for smeared boundary state (left) and its lattice discretization (right) as a matrix product state. On the right, each vertical column of lattice sites represents a matrix. The horizontal lines represent contractions between these in the internal space, and the vertical dangling bonds label the physical degrees of freedom.

From this point of view it is therefore natural to regard (4) as a variational *ansatz*, with $\tau$ and the choice of boundary state $|\mathscr{B}\rangle$ as variational parameters.

In this paper we explore this idea further and show that this program can be carried through explicitly for the $A_m$ (diagonal) series of unitary minimal 2d CFTs. It should be extendable to the other non-diagonal minimal models, and in principle to other rational 2d CFTs, and indeed to higher dimensional theories if enough information is available about the CFT.

More specifically, given a set of physical conformal boundary states $\{|a\rangle\}$, (whose definition is recalled in Sec. 2) we take as a variational ground state

$$|\{\alpha_a\}, \{\tau_a\}\rangle = \sum_a \alpha_a \, e^{-\tau_a \hat{H}_{CFT}} |a\rangle \, , \tag{5}$$

and compute the variational energy per unit volume

$$\lim_{L \to \infty} \frac{1}{L^D} \frac{\langle \{\alpha_a\}, \{\tau_a\} | \hat{H}_{CFT} + \sum_j \lambda_j \int \hat{\Phi}_j(x) d^{d-1}x | \{\alpha_a\}, \{\tau_a\} \rangle}{\langle \{\alpha_a\}, \{\tau_a\} | \{\alpha_a\}, \{\tau_a\} \rangle} \, , \tag{6}$$

minimizing this with respect to the $\{\alpha_a\}$ and $\{\tau_a\}$.

An important general consequence of the analysis is that, in the limit $L \to \infty$, the minimizing states are always purely physical, that is all but one $\{\alpha_a\}$ vanishes. This is because both $H_{CFT}$ and the perturbing operators turn out to be diagonal in this basis. This is reassuring, as in principle the minimizers could be non-physical linear combinations of these, for example the Ishibashi states in 1+1 dimensions.

Specializing now to the case of 1+1 dimensions, for the minimal models, the precise values of these diagonal matrix elements are related to the elements of the modular $S$-matrix of the CFT, and, with these in hand, it is straightforward, for a fixed set of couplings $\{\lambda_j\}$ to determine which values of $\tau_a$ and $a$ minimize the variational energy, and thus to map out a rudimentary phase diagram of the theory in the vicinity of the CFT.

It then turns out that although this approach yields correct results in some aspects, for example in determining which combination of bulk couplings $\{\lambda_j\}$ best matches a given boundary state $a$, it is not capable of reproducing some of the finer details of the phase boundaries between different states $a$. In this approximation these are always first-order, and 'massless' RG flows to other non-trivial CFTs are not properly accounted for. This can be seen as a limitation of the particular trial state, which could be remedied by including other operators acting on the boundary state, but at the cost of the loss of analytic tractability.

However, an amusing side result of the analysis is that matrix elements of primary bulk operators $\hat{\Phi}_j$ between Ishibashi states $\langle\langle i|, |k\rangle\rangle$ (which are boundary states within a single Virasoro module) are simply proportional to the fusion rule coefficients:

$$\langle\langle i| e^{-\tau H} \hat{\Phi}_j \, e^{-\tau H} |k\rangle\rangle \propto N_{ijk} \, . \tag{7}$$

This is a consequence of the Verlinde formula [6], and to our knowledge has not been previously observed. Although this matrix element should be proportional to the OPE coefficient $c_{ijk}$ which governs the matrix element $\langle i|\hat{\Phi}_j|k\rangle$ between highest weight states, it is rather surprising that the contributions of all the descendent states should conspire to give the integer-valued fusion rule coefficient.

The outline of this paper is as follows. In Sec. 2 we set up the formalism and prove some general results. In Sec. 3 we apply this to the case of the diagonal minimal models, with the $A_3$ and $A_4$ cases as specific examples, and finally in Sec. 4 give a summary and some further remarks.

After this paper was completed, I was made aware of Ref. [7], in which similar ideas are explored. However that paper is based on comparing ratios of overlaps between different

boundary states and numerical approximations to the exact ground state of the deformed theory, rather than the variational method adopted here. The overlap method is shown to work well for the case of the perturbed Ising model, but is computationally more intensive.

## 2   General formalism

As described in the Introduction, we consider a $d$ $(= D + 1)$-dimensional CFT perturbed by its bulk primary operators $\{\Phi_j\}$ with coupling constants $\{\lambda_j\}$, so the hamiltonian is:

$$\hat{H} = \hat{H}_{CFT} + \sum_j \lambda_j \int \hat{\Phi}_j(x) d^D x \,. \tag{8}$$

The theory is quantized on a spatial torus of volume $L^D$, where $L$ is much larger than any other scale in the theory. We assume for simplicity that the $\{\Phi_j\}$ are all scalars and that they have their CFT normalization

$$\langle \hat{\Phi}_j(x) \hat{\Phi}_j(0) \rangle_{CFT} = |x|^{-2\Delta_j} \,, \tag{9}$$

where $\Delta_j$ is the scaling dimension of $\Phi_j$.

   Although we are interested in relevant perturbations with $\Delta_j < d$, these will in general lead to a finite number of primitive UV divergences up to some finite order in the couplings (as for a super-renormalizable deformation of a free theory), in particular in the ground state energy which we are trying to approximate. These divergences may be subtracted by adding a finite number of counter-terms to $\hat{H}$ determined the OPEs of the $\{\Phi_j\}$. We assume this has been done. For example, if $2\Delta_j \geq d$ there is a UV divergence in the ground state energy at $O(\lambda_j^2)$. This is subtracted by a term in $\hat{H}$ proportional to the unit operator. This does not affect the variational procedure in general. The case $2\Delta_j = d$ is special and leads to a logarithmic anomaly in the energy. This will be discussed for the 2d Ising model in Sec. 3.1.1.

   Conformal boundary states $|\mathscr{B}\rangle$ are defined by the condition

$$\hat{T}_{0k}(x)|\mathscr{B}\rangle = 0, \quad (k = 1, \dots, D), \tag{10}$$

where $\hat{T}_{ij}$ is the energy-momentum tensor of the CFT. That is, they are annihilated by the momentum density operator, and so are invariant under local time reparametrizations. (For boundaries with a space-like normal there is no energy flow across the boundary.)

   Although in higher dimensions these states, and their classification, are poorly understood except for free or weakly coupled CFTs, in 2d much more is known [8,9]. The Hilbert space is acted on by two copies $(\mathscr{V} \otimes \overline{\mathscr{V}})$ of the Virasoro algebra, generated by

$$\hat{L}_n = \frac{L}{2\pi} \int e^{2\pi n i x / L} \hat{T}(x) dx, \quad \hat{\overline{L}}_n = \frac{L}{2\pi} \int e^{-2\pi n i x / L} \hat{\overline{T}}(x) dx, \tag{11}$$

where, as usual, $T \equiv T_{zz} = -T_{00} + T_{11} - 2iT_{01}$ and $\overline{T} \equiv T_{\bar{z}\bar{z}} = -T_{00} + T_{01} + 2iT_{01}$, in euclidean signature. It is spanned by states $|i, N\rangle \otimes \overline{|i', N'\rangle}$, where $N$ labels the states of a module of $\mathscr{V}$ with highest weight state labelled by $i$, and similarly for $\overline{\mathscr{V}}$. For CFTs with central charge $c \geq 1$, this is a Virasoro module, while for the minimal models with $c < 1$ it is a Kac module with the null states projected out.

   The condition (10) then corresponds to

$$\big(\hat{T}(x) - \hat{\overline{T}}(x)\big)|\mathscr{B}\rangle = 0 \,. \tag{12}$$

In terms of the Virasoro generators this becomes

$$\big(\hat{L}_n - \hat{\overline{L}}_{-n}\big)|\mathscr{B}\rangle = 0 \,, \tag{13}$$

whose solution is the span of the Ishibashi states

$$|i\rangle\rangle = \sum_N |i, N\rangle \otimes \overline{|i, N\rangle}, \tag{14}$$

where the sum is over all the orthonormalized states in the module.

However, the Ishibashi states are not physical, in the sense that, when they are chosen as boundary states on the opposite edges $x_0 = \pm\tau$ of an annulus, the partition function $Z = \mathrm{Tr}\, e^{-L\hat{H}'}$ evaluated in terms of the generator $\hat{H}'$ of translations around the annulus does not have the form of a sum over eigenstates with non-negative integer coefficients, as it must if periodic spatial boundary conditions are imposed. For the diagonal minimal $A_m$ models, the physical states which do have this property are linear combinations of the Ishibashi states

$$|a\rangle = \sum_j \frac{S_a^i}{(S_0^i)^{1/2}} |i\rangle\rangle, \tag{15}$$

where $S_k^i$ is the matrix by which the Virasoro characters transform under modular transformations. The multiplicities of the eigenstates $j$ of $\hat{H}'$ which do propagate are then given by the fusion rule coefficients $N_{ab}^j$. In particular the vacuum state propagates only if $a = b$, that is $N_{ab}^0 = \delta_{ab}$. While similar results are available for the non-diagonal minimal models, wider results for general CFTs are not available, which is why we mainly restrict to the $A_m$ models in explicit calculations.

In higher dimensions, the boundary states satisfying (10) also form a linear space, and we assume that the physical states may be identified analogously. Consider the partition function in the slab $\mathbb{T}^D \times \{-\tau, \tau\}$ (where $\mathbb{T}^D$ is a $D$-dimensional torus of volume $L^D$) with boundary states $|a\rangle$, $|b\rangle$ at $x_0 = \pm\tau$:

$$Z_{ab} = \langle b|e^{-2\tau\hat{H}_{CFT}}|a\rangle, \tag{16}$$

(where $\hat{H}_{CFT}$ is the generator of translations in $x_0$) and, similarly to the 2d case, demand that when evaluated by quantizing in one of the spatial directions, it has the form of a trace over intermediate states whose energies all scale like $\tau^{-1}$. However this is difficult to implement since this spectrum on the torus is not related to the conformal spectrum for $d > 2$.

In fact, we shall need only a weaker condition: that physical states $\{a, b, \ldots\}$ should satisfy

$$Z_{ab}/(Z_{aa}Z_{bb})^{1/2} = O\big(e^{-\mathrm{const.}(L/2\tau)^D}\big) \qquad \text{for } L/\tau \to \infty. \tag{17}$$

This may be understood as follows: when the boundary conditions are the same, we expect that

$$Z_{aa} \sim e^{\sigma_a(L/2\tau)^D}, \tag{18}$$

where the exponent is (minus) the Casimir energy of a system between two identical plates. In this geometry this is always attractive, thus $\sigma_a > 0$. It must scale as $L^D$, and, since the boundary conditions and the bulk theory are scale-invariant, also as $\tau_a^{-D}$. The quantity $-\sigma_a L^{D-1}/(2\tau)^D$ is the ground state energy of the generator of translations around one of the spatial cycles of the torus. On the other hand the exponent on the right hand side of (17) is the gap to the lowest-energy state in the sector with $ab$ boundary conditions. It is the interfacial energy from the point of view of $d$-dimensional classical statistical mechanics.

Thus the physical boundary states may in principle be determined by diagonalizing the partition functions in the slab in the limit $L \gg \tau$. We assume that this has been done.

As discussed in the introduction, we use as variational states for the ground state of the perturbed hamiltonian (8), the ansatz

$$|\{\alpha_a\}, \{\tau_a\}\rangle = \sum_a \alpha_a\, e^{-\tau_a\hat{H}_{CFT}}|a\rangle. \tag{19}$$

We first discuss the inner product of these states

$$\langle a|e^{-\tau_a H_{CFT}} e^{-\tau_b H_{CFT}}|b\rangle. \tag{20}$$

This is the partition function $Z_{ab}$ in slab of width $\tau_a + \tau_b$ with boundary conditions $a, b$ on opposite faces. As discussed above, for physical boundary states in the limit $L \gg \tau_a + \tau_b$

$$Z_{ab} \sim \delta_{ab} e^{\sigma_a (L/(2\tau_a))^D}. \tag{21}$$

The matrix elements of the unperturbed hamiltonian $\hat{H}_{CFT}$ are, for the same reason, diagonal in this basis as long as $L \gg \tau_{a,b}$, and may be found by differentiating (21)

$$\langle a|\hat{H}_{CFT} e^{-2\tau_a \hat{H}_{CFT}}|a\rangle \sim \delta_{ab} \frac{D\sigma_a L^D}{(2\tau_a)^{D+1}} e^{\sigma_a (L/(2\tau_a))^D}. \tag{22}$$

Finally we need the matrix elements of the perturbation

$$\langle a|e^{-\tau_a \hat{H}_{CFT}} \hat{\Phi}_j(x) e^{-\tau_b \hat{H}_{CFT}}|b\rangle, \tag{23}$$

which is a one-point function in the slab. Once again, if we evaluate this by inserting a complete set of eigenstates of a generator of translations around the torus, this is dominated by its ground state if $L \gg \tau_{a,b}$, but this contributes only if $a = b$. So the perturbation is also diagonal in the basis of physical states (but not in the Ishibashi basis: see Sec. 3.3).

When $a = b$ the one-point functions in the mid-plane of the slab have the form

$$\langle \Phi_j(x)\rangle = \frac{A_a^j}{(2\tau_a)^{\Delta_j}}, \tag{24}$$

where the amplitudes $A_a^j$ are universal given the normalization (9) of the operator.

Since the perturbed hamiltonian is diagonal in the physical basis of variational states, the problem becomes much simpler: for each $a$ we should minimize the variational energy per unit volume

$$E_a = \frac{D\sigma_a}{(2\tau_a)^{D+1}} + \sum_j \lambda_j \frac{A_a^j}{(2\tau_a)^{\Delta_j}}, \tag{25}$$

with respect to $\tau_a$, and then choose the $a$ which gives the absolute minimum.

Note that having found this minimum $a$ for a particular set of couplings $\{\lambda_j\}$, since $E_a$ transforms multiplicatively under

$$\lambda_j \to e^{(D+1-\Delta_j)\ell}\lambda_j, \quad \tau_a \to e^{-\ell}\tau_a, \quad E_a \to e^{(D+1)\ell}E_a, \tag{26}$$

the absolute minimum will occur for the same value of $a$ along an RG trajectory. This is reassuring, since each point on the trajectory should be described by the same massive QFT up to a rescaling of the mass, which is proportional to $1/\tau_a^{\min}$.

Since the $\{\Phi_j\}$ are relevant, $\Delta_j < D+1$, so that the behavior of $E_a$ as $\tau_a \to 0$ (but still $\gg \epsilon$) is dominated by the first term and is positive if $\sigma_a > 0$ (which corresponds to the physically reasonable case of an attractive Casimir force.) As $\tau_a \to \infty$ it approaches zero, dominated by the term with smallest $\Delta_j$ and $\lambda_j \neq 0$. If the sign is negative this implies that $E_a$ has a negative minimum at some finite value of $\tau_a$. At least for the 2d minimal models we can show that there is always some $a$ for which $\lambda_j A_a^j < 0$, so this minimum always exists.

### 2.1 Trace of the energy-momentum tensor

We may infer a general result about the trace $\langle \Theta \rangle = \langle T_i^i \rangle$ of the energy-momentum tensor in the perturbed theory as approximated by this method. For given set of relevant perturbations $\{\lambda_j\}$ this is given by the response of the action to a scale transformation

$$\Theta(x) = -\sum_j (D + 1 - \Delta_j)\lambda_j \Phi_j(x). \tag{27}$$

Differentiating (25) we see that at the minimum

$$\frac{(D+1)D\sigma_a}{(2\tau_a)^{D+1}} + \sum_j \Delta_j \lambda_j \langle \Phi_j \rangle = 0, \tag{28}$$

and so

$$E = -(D+1)^{-1} \sum_j \Delta_j \lambda_j \langle \Phi_j \rangle + \sum_j \lambda_j \langle \Phi_j \rangle = -(D+1)^{-1}\langle \Theta \rangle. \tag{29}$$

Once again, this is reassuring, as we expect that in the ground state of a relativistic theory $\langle T_{00} \rangle = -\langle T_{kk} \rangle$ for $k \neq 0$, and so

$$\langle \Theta \rangle = -\langle T_{00} \rangle + \sum_{k=1}^D \langle T_{kk} \rangle = -(D+1)\langle T_{00} \rangle = -(D+1)E. \tag{30}$$

The variational method therefore gives a lower bound on $\langle \Theta \rangle$.

## 3 2d minimal models

We now specialize to the case of the 2d $A_m$ minimal models.

In 2d, the Casimir amplitude $\sigma_a = \pi c/24$, independent of $a$, where $c$ is the central charge. When $a = b$ the expectation values of the one-point functions in a long strip of width $2\tau_a$ may be found by a conformal mapping from the half-plane to have the form

$$\langle \Phi_j(x, \tau) \rangle_{\text{strip}} = \frac{\widetilde{A}_a^j}{((2\tau_a/\pi)\sin(\pi\tau/2\tau_a))^{\Delta_j}}, \tag{31}$$

where the amplitude governs the behavior of the one-point function in the upper half-plane $y > 0$ with boundary condition $a$ on the real axis:

$$\langle \Phi_j(y) \rangle_{\text{half-plane}} = \frac{\widetilde{A}_a^j}{y^{\Delta_j}}. \tag{32}$$

In (31) we should set $\tau = \tau_a$, whence we read off that $A_a^j = \pi^{\Delta_j}\widetilde{A}_a^j$.

If the operator $\Phi_j$ has its standard CFT normalization (9), the amplitudes $\widetilde{A}_a^j$ are universal. In [10] they were computed in terms of the overlap between the boundary state $|a\rangle$ and the highest weight state $|j\rangle$ corresponding to the primary operator $\Phi_j$:

$$\widetilde{A}_a^j = \frac{\langle j|a \rangle}{\langle 0|a \rangle}. \tag{33}$$

This follows by conformally mapping the upper half plane to a semi-infinite cylinder $x > 0$ with a boundary condition $a$ at $x = 0$, and comparing the result for $x \to \infty$ with the result of inserting a complete set of eigenstates of the generator of translations along the cylinder.

For the $A_m$ minimal models, inserting the expression (15) for $|a\rangle$ we then find [10]

$$\widetilde{A}_a^j = \frac{S_a^j}{S_a^0} \left( \frac{S_0^0}{S_0^j} \right)^{1/2}. \tag{34}$$

To summarize, the variational energy (25) in this case is given by

$$E_a = \frac{\pi c}{24(2\tau_a)^2} + \sum_j \lambda_j \frac{S_a^j}{S_a^0} \left( \frac{S_0^0}{S_0^j} \right)^{1/2} \frac{\pi^{\Delta_j}}{(2\tau_a)^{\Delta_j}}. \tag{35}$$

It is useful to rescale the couplings by positive constants $\tilde{\lambda}_j = \pi^{\Delta_j} (S_0^0/S_0^j)^{1/2} \lambda_j$ so that this simplifies to

$$E_a = \frac{\pi c}{24(2\tau_a)^2} + \sum_{j \neq 0} \frac{S_a^j}{S_a^0} \frac{\tilde{\lambda}_j}{(2\tau_a)^{\Delta_j}}. \tag{36}$$

Note that that sum over $j$ excludes $j = 0$ which corresponds to adding the unit operator and therefore a constant shift in the energy.

There are two general statements which follow from the fact that $S$ is a symmetric orthogonal matrix, and that the elements $S_0^j$ are all positive.

First, since all its rows are orthogonal and non-zero it follows that, for $j \neq 0$, some of the elements $S_a^j$ are positive and some negative. Therefore, if $j^*$ corresponds to the smallest value of $\Delta_j$ such that $\lambda_j \neq 0$, and therefore dominates the behavior of $E_a$ as $\tau_a \to \infty$, no matter what the sign of $\lambda_{j^*}$ we may always find at least one $a$ such that $\lambda_{j^*} S_a^j < 0$, and so $E_a$ approaches zero from below. Since $E_a \to +\infty$ as $\tau_a \to 0$, this implies that, for these $a$, $E_a$ has a negative minimum at finite $\tau_a$, corresponding to a finite correlation length. This rules out the possibility that this variational ansatz can describe massless flows to another non-trivial CFT.

Second, we may ask whether there is a combination of couplings $\{\lambda_j\}$ which will lead to a prescribed $b$ as overall minimum. The answer is affirmative. For, suppose we choose

$$\tilde{\lambda}_j = -g(2\mu)^{\Delta_j - 2} S_b^j, \tag{37}$$

where $g$ is a positive constant and $\mu$ is some fixed scale $> \epsilon$. Then the second term in (25) is, when $\tau_a = \mu$,

$$-\frac{g}{S_a^0 \mu^2} \sum_{j \neq 0} S_a^j S_b^j = -\frac{g}{S_a^0 \mu^2} \left( \delta_{ab} - S_a^0 S_b^0 \right). \tag{38}$$

Since $0 < S_a^0 < 1$, this is $< 0$ if $a = b$ and $> 0$ otherwise. Thus, at this scale, the boundary state $b$ will correspond to the lowest trial energy[2]. Note that (37) implies including some irrelevant couplings in the mix of deformations.

Further results depend on the detailed form of the modular $S$-matrix for the $A_m$ models. In particular, we may ask what happens if a single $\lambda_j$ is non-zero. Depending on whether $\lambda_j > 0$ or $< 0$, we have to determine which value of $a$ minimizes (maximizes) the ratio $S_a^j/S_a^0$.

Label the positions of the bulk operators in the Kac table by $j = (r, s)$, with $1 \leq r \leq m - 1$ and $1 \leq s \leq m$, and $(r, s)$ identified with $(m - r, m + 1 - s)$. The label $j = 0$ corresponds to $(r, s) = (1, 1)$. Similarly label the boundary states $a$ by $(\alpha, \beta)$.

The $A_m$ minimal series of CFTs is conjectured to be the scaling limit of the critical lattice RSOS $A_m$ models [11, 12]. These models are defined on a square lattice. At each node $R$ there

---

[2]This does not rule out the possibility that some other $E_a$ might come lower than this at some other scale, but in practice this does not seem to happen.

is an integer-valued height variable $h(R)$ satisfying $1 \le h(R) \le m$, with the RSOS constraint that $|h(R) - h(R')| = 1$ if $R$ and $R'$ are nearest neighbors. The heights may be thought of as living at the nodes of the Dynkin diagram $A_m$, so each configuration is a many-to-one embedding of the diagram into the square lattice. The critical Boltzmann weights of the lattice model are specified in terms of the elements $s_h^0(m)$ of the Perron-Frobenius eigenvector corresponding to the largest eigenvalue of the adjacency matrix of $A_m$. The general eigenvector has the form

$$s_h^j(m) \propto \sin \frac{\pi j h}{m+1}. \tag{39}$$

The microscopic interpretation of the conformal boundary states (15) for these models has been given in [13, 14]. The simplest boundary states are when the boundary lies at 45° to the principal lattice vectors, and the heights on the boundary are all fixed to the same particular value $h$, say. These have been identified with the conformal boundary conditions with the Kac labels $(\alpha, \beta) = (1, h)$. The second simplest type of microscopic boundary condition is when the boundary heights are fixed to $h$ and on the neighboring diagonal they are fixed to $h+1$. These have been identified with $(\alpha, \beta) = (h, 1)$. In [14] a complete set of microscopic boundary conditions was identified for each Kac label $(\alpha, \beta)$ but these become increasingly complicated. In general the microscopic boundary states corresponding to labels near the center of the Kac table are increasingly disordered.

The ratios of elements of the modular $S$-matrix for the diagonal $A_m$ models are

$$\frac{S_a^j}{S_a^0} = \frac{S_{\alpha,\beta}^{r,s}}{S_{\alpha,\beta}^{1,1}} = (-1)^{(r+s)(\alpha+\beta)} \frac{\sin \frac{\pi r \alpha}{m}}{\sin \frac{\pi \alpha}{m}} \frac{\sin \frac{\pi s \beta}{m+1}}{\sin \frac{\pi \beta}{m+1}} = (-1)^{(r+s)(\alpha+\beta)} \frac{s_\alpha^r(m-1)}{s_\alpha^1(m-1)} \frac{s_\beta^s(m)}{s_\beta^1(m)}. \tag{40}$$

Locating the global maximum and minimum of this expression for general $(r, s)$ is simplified by the fact that, for fixed $(r, s)$, it factorizes into expressions depending only on $\alpha$ and $\beta$ respectively. Thus we can restrict to the four possible products of the maximum and minimum of each factor, and compare these values. In each factor the numerator is an oscillating function which is modulated by the positive denominator, which itself has minima at $\alpha = 1, m-1$ (and $\beta = 1, m$), which for the lattice $A_m$ models correspond to the most ordered states.

The most relevant bulk operator corresponds to $(r, s) = (2, 2)$, when (40) becomes

$$4 \cos \frac{\pi \alpha}{m} \cos \frac{\pi \beta}{m+1}. \tag{41}$$

The extrema of each factor are at $\alpha = 1, m-1$ and $\beta = 1, m$. Thus for $\lambda_{2,2} > 0$ the minimum energy corresponds to $(\alpha, \beta) = (1, m) = (m-1, 1)$, and, for $\lambda_{2,2} < 0$, $(\alpha, \beta) = (1, 1) = (m-1, m)$. These correspond to the most ordered states, at the ends of the Dynkin diagram. This is to be expected as, in the Landau-Ginzburg correspondence, $\Phi_{2,2}$ is the most relevant $Z_2$ symmetry breaking operator.

Similarly, the most relevant $Z_2$ even operator is $\Phi_{3,3}$, when (40) becomes

$$(2 \cos \frac{2\pi \alpha}{m} + 1)(2 \cos \frac{2\pi \beta}{m+1} + 1). \tag{42}$$

If $m$ is even, the first factor varies between $2 \cos \frac{2\pi}{m} + 1$ at $\alpha = 1, m-1$, and $-1$ at $\alpha = \frac{1}{2}m$, and the second factor varies between $2 \cos \frac{2\pi}{m+1} + 1$ at $\beta = 1, m$, and $2 \cos \frac{\pi m}{m+1} + 1$ at $\beta = \frac{1}{2}m, \frac{1}{2}m + 1$. Thus for $\lambda_{3,3} < 0$ there are degenerate minima for $(\alpha, \beta) = (1, 1) = (m-1, m)$ and $(\alpha, \beta) = (1, m) = (m-1, 1)$ (the most ordered states, which break the $Z_2$ symmetry.).

On the other hand for $\lambda_{3,3} > 0$ we need to compare the quantities

$$(-1)(2 \cos \frac{2\pi}{m+1} + 1), \quad (2 \cos \frac{\pi m}{m+1} + 1)(2 \cos \frac{2\pi}{m} + 1). \tag{43}$$

Numerically, the first is more negative, so the minimum energy in this case corresponds to $\alpha = \frac{1}{2}m$, $\beta = 1, m$. These are $Z_2$-symmetric states. For odd $m$ the same story holds, with $\alpha$ and $\beta$ interchanged.

Another interesting special case is $\Phi_{1,3}$. This is a perturbation which, with the correct sign, is supposed to flow to the $A_{m-1}$ minimal CFT, and for the other sign to a state with large degeneracy [11]. As we have seen, such massless flows cannot be accounted for within this set of trial states. In this case (40) simplifies to

$$2\cos\frac{2\pi\beta}{m+1}+1\,,\tag{44}$$

independent of $\alpha$. Depending on the sign of the coupling, this picks out the boundary states either with $\beta = 1, m$ or with $\beta \approx \frac{1}{2}m$. In both cases, however, there is an $(m-1)$-fold degeneracy of candidate ground states. This reflects a flow towards a true first-order transition, as expected for one sign of the coupling [11], or the best attempt of this approximation to reproduce the critical point of the $A_{m-1}$ model, as expected for the other sign. This is an important check on the effectiveness of our approach.

These somewhat cryptic general remarks are best illustrated with some simple examples.

## 3.1 The Ising model

This corresponds to $A_3$. The perturbed hamiltonian is

$$\widehat{H} = \widehat{H}_{CFT} + t\int \hat{\epsilon}\,dx + h\int \hat{\sigma}\,dx\,,\tag{45}$$

where $\varepsilon = \Phi_{2,1} = \Phi_{1,3}$ and $\sigma = \Phi_{1,2} = \Phi_{2,2}$ are the energy density and magnetization operators respectively.

In this case, the bulk operators are $\{\Phi_j\} = (1,\epsilon,\sigma)$, and the boundary states in the same labeling are $(+,-,f)$, corresponding to fixed(+), fixed($-$) and free boundary conditions on the Ising spins. The $S$-matrix in this ordering of the basis is

$$S = \begin{pmatrix} \frac{1}{2} & \frac{1}{2} & \frac{1}{\sqrt{2}} \\ \frac{1}{2} & \frac{1}{2} & -\frac{1}{\sqrt{2}} \\ \frac{1}{\sqrt{2}} & -\frac{1}{\sqrt{2}} & 0 \end{pmatrix}.\tag{46}$$

After rescaling the couplings as above, we find that

$$E_+ = \frac{\pi}{48(2\tau_+)^2} + \frac{t}{2\tau_+} + \sqrt{2}\frac{h}{(2\tau_+)^{1/8}}\,,\tag{47}$$

$$E_- = \frac{\pi}{48(2\tau_-)^2} + \frac{t}{2\tau_-} - \sqrt{2}\frac{h}{(2\tau_-)^{1/8}}\,,\tag{48}$$

$$E_f = \frac{\pi}{48(2\tau_f)^2} - \frac{t}{2\tau_f}\,.\tag{49}$$

For $h = 0$, $t > 0$, corresponding to the disordered state, it is clear that the minimizer is $E_f$. For the opposite sign of $t$ with $h > 0$, the minimizer is $E_-$, corresponding to negative magnetization (recall the definition of the sign of the couplings in (45)), and vice versa. As $h \to 0$ from either side with $t < 0$, we remain in one or the other of these states, corresponding to spontaneous symmetry breaking.

However, for $t > 0$ and $0 < h \ll t^{15/8}$ there is a problem. The minimum of $E_-$ is found by balancing the last two terms in (48), and therefore occurs when $\tau_- = O((t/h)^{7/8})$ at a value $E_- = -O(h^{8/7}/t^{1/7})$. On the other hand the minimum of $E_f = -O(t^2)$ is much lower in this

limit. This would suggest, incorrectly, that the magnetization is zero in the ground state. As we increase the ratio $h/t^{15/8}$, eventually these levels cross, but there is no reason for $\tau_-$ and $\tau_f$ to be equal at this point.

This appears to be an inherent problem of using a variational ansatz which is not sufficiently complex. It could presumably be overcome by using a trial state of the form

$$e^{-\tau \hat{H}_{CFT}} e^{-h_s \int \hat{\sigma}_s dx} |f\rangle, \tag{50}$$

where $\hat{\sigma}_s$ is the boundary magnetization coupling to a boundary magnetic field $h_s$, at the cost of loss of analytic tractability.

### 3.1.1 Logarithmic anomaly

When $h = 0$ it follows from (47,48,49) that the minimum energy scales like $t^2$. Yet it has been known since Onsager that the correct behavior is $t^2 \log t$. The origin of this logarithmic anomaly is a cancellation between the scaling term $t^{2/(2-\Delta_\varepsilon)}$ and the analytic background $\propto t^2$, both of which occur with amplitudes which diverge as $\Delta_\varepsilon \to 1$. This may be accounted for within the variational approach by adding a counter-term as before, proportional to the space-time integral of the 2-point function, which will now also have a logarithmic dependence on the IR cutoff $\tau$. Thus, for example, (49) becomes

$$E_f = \frac{\pi}{48(2\tau_f)^2} - \frac{t}{2\tau_f} - A t^2 \log(\tau_f/\epsilon), \tag{51}$$

where $\epsilon$ is the short-distance cutoff and $A$ is a (calculable) $O(1)$ constant. The minimum still occurs at $\tau_f \sim t^{-1}$, but the last term now contributes the desired logarithm at the minimum.

### 3.2 The tricritical Ising model

This corresponds to the $A_4$ lattice model with heights $h(R) \in \{1, 2, 3, 4\}$. The RSOS condition means that $h$ is even on even sites odd $s$ on odd sites, or vice versa.

In the Landau-Ginzburg picture it corresponds to a scalar field $\phi$ with a $\phi^6$ interaction, and a $Z_2$ symmetry under $\phi \to -\phi$. Note that in the lattice model this $Z_2$ symmetry is implemented by reflecting the Dynkin diagram *and* a sublattice shift. The Kac table with bulk operators labelled by Landau-Ginzburg is shown in Fig. 2. Note that odd $r$ is $Z_2$ odd and vice versa.

However another model in the same universality class is the spin-1 (Blume-Capel) Ising model, which may be thought of as an Ising model with vacancies.

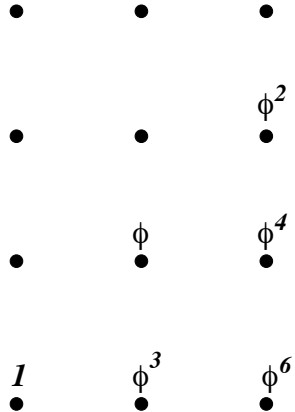

Figure 2: Landau-Ginzburg assignment of bulk operators in the $A_4$ Kac table.

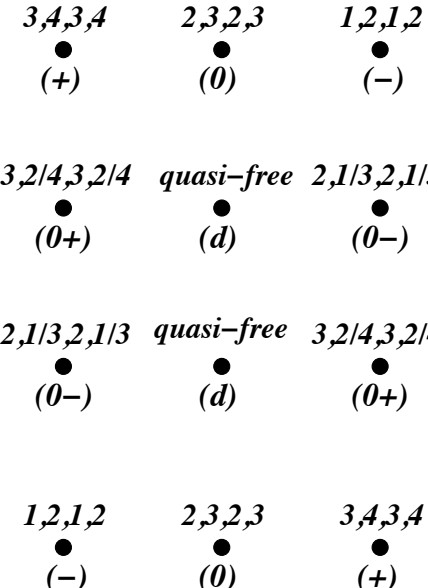

Figure 3: Correspondence between boundary conditions in lattice models and Kac labels of conformal boundary states: in the $A_4$ model according to Ref. [14] (upper labels), and in the Blume-Capel model, according to Ref. [16] (lower labels).

The usually accepted phase diagram and RG flows of the tricritical Ising model near the tricritical fixed point are quite complex. (See for example Fig. 4.2 of [17].) In the $Z_2$-even sector, turning on the most relevant operator $\Phi_{3,3} \sim \phi^2$ gives flows either to the high-temperature disordered phase, or to the 2 coexisting low-temperature ordered phases. Turning on the $\Phi_{1,3} \sim \phi^4$ operator gives flows either to a first-order transition between these ordered phases and a disordered phase with vacancies, or to the $A_3$ Ising fixed point.

As for the Ising model, turning on the $\Phi_{2,2} \sim \phi$ operator leads to broken-symmetry phases. However, at low temperatures there may be coexistence between two such phases with different densities of vacancies. These persist to finite temperature, giving 'wings' in the phase diagram which end in lines of Ising-like transitions. These lines meet in the tricritical point and correspond to flows generated by the non-leading but relevant $Z_2$-odd operator $\Phi_{2,1} \sim \phi^3$.

According to Behrend and Pearce [14], the labelling of the boundary states in the $A_4$ lattice model is as shown in Fig. 3. On the same diagram we have indicated their interpretation in the Blume-Capel model, due to Chim [15] and Affleck [16], which is perhaps more intuitive. Here $(\pm)$ label totally ordered states, (0) is a vacancy-rich state, and $(0\pm)$ are partially ordered states. ($d$) is a multicritical point separating these in the boundary RG flows [16]. Note that the $\alpha = 2$ states are $Z_2$ even while the $Z_2$ symmetry interchanges $\alpha = 1$ and $\alpha = 3$ (keeping $\beta$ the same.)

Let us see how well the variational approach reproduces this picture. According to the earlier analysis, turning on the $\Phi_{2,2}$ operator corresponds to the boundary states at the corners of the Kac table in Fig. 3.2. These are the most ordered states.

Again, turning on the $\Phi_{3,3}$ perturbation corresponds to the boundary states which extremize $\left(2\cos(\pi\alpha/2)+1\right)\left(2\cos(2\pi\beta/5)+1\right)$. This gives $\beta = 1, 4$, and, depending on the sign of the coupling, either $\alpha = 2$ or $\alpha = 1, 4$. These correspond to the disordered and ordered Ising-like phases, respectively, as expected.

Turning on $\Phi_{1,3}$ corresponds to maximizing only the second factor $\left(2\cos(2\pi\beta/5)+1\right)$, and so, depending on the sign of the coupling gives either $\beta = 1, 4$, corresponding to coexistence between these Ising-like phases (instead of a second order critical point as it should), or

$\beta = 2, 3$ coexistence between partially ordered phases and a disordered, vacancy-rich phase.

Turning on $\Phi_{2,1}$, on the other hand, corresponds to extremizing $(-1)^{\alpha+\beta} \cos(\pi\alpha/4)$. For one sign of the coupling we get coexistence between the strongly ordered phase $(-) = (1, 2, 1, 2)$ and the partially ordered phase $(0+) = (3, 2/4, 3, 2/4)$, and for the other sign we get coexistence between their $Z_2$ partners. This is once again in general agreement with the wings of the phase diagram, except that the approximation suggests a first-order rather than an Ising-like continuous transition.

We conclude that for this model the boundary states roughly reproduce the expected RG flows when a single relevant operator is turned on, with the exception that flows to non-trivial CFTs are approximated by first-order rather than continuous transitions.

## 3.3 Matrix elements between Ishibashi states and the fusion rules

As an aside, we mention a curiosity which follows from the result (34) for the matrix element of a bulk primary field between physical states in the limit $L \rightarrow \infty$:

$$\langle a | e^{-\tau \hat{H}} \hat{\Phi}_j e^{-\tau \hat{H}} | b \rangle = \delta_{ab} \left( \frac{\pi}{2\tau} \right)^{\Delta_j} \frac{S_a^j}{S_a^0} \left( \frac{S_0^0}{S_0^j} \right)^{1/2} \tag{52}$$

and the definition of these states in terms of the Ishibashi states (15), which, on inverting becomes:

$$|i\rangle\rangle = \sum_a S_a^i (S_0^i)^{1/2} |a\rangle. \tag{53}$$

Hence

$$\langle\langle i | e^{-\tau \hat{H}} \hat{\Phi}_j e^{-\tau \hat{H}} | k \rangle\rangle = \left( \frac{\pi}{2\tau} \right)^{\Delta_j} \left( \frac{S_0^0}{S_0^j} \right)^{1/2} (S_0^i)^{1/2} (S_0^k)^{1/2} \sum_a \frac{S_a^i S_a^j S_a^k}{S_a^0}. \tag{54}$$

We recognize the sum over $a$ as the Verlinde formula [6] for the fusion rule coefficient $N_{ijk}$. Taking into account the normalization of the states, we have

$$\frac{\langle\langle i | e^{-\tau \hat{H}} \hat{\Phi}_j e^{-\tau \hat{H}} | k \rangle\rangle}{\left( \langle\langle i | e^{-2\tau \hat{H}} | i \rangle\rangle \langle\langle k | e^{-2\tau \hat{H}} | k \rangle\rangle \right)^{1/2}} = \left( \frac{\pi}{2\tau} \right)^{\Delta_j} \left( \frac{S_0^0}{S_0^j} \right)^{1/2} N_{ijk}. \tag{55}$$

Note that the first factor could be absorbed into a redefinition of the normalization of $\hat{\Phi}_j$.

This result is somewhat surprising, and, to our knowledge, has not been noticed before. If we insert the definition (14) of the Ishibashi states into the numerator, the leading term for $\tau \gg L$ is proportional to the OPE coefficient $c_{ijk}$, which certainly vanishes whenever $N_{ijk}$ does. The contributions of all the descendent states are all proportional to $c_{ijk}$, with coefficients which could, in principle, be computed from the Virasoro algebra. However, it is remarkable that they all conspire to sum to the integer-valued fusion rule coefficient $N_{ijk}$.

If there were an independent way of establishing (55) this would give an alternative derivation of the Verlinde formula. It would be interesting to see whether this result extends to non-diagonal minimal models and to other rational CFTs. Although it has been derived here only for the Virasoro minimal models, there seems to be no obstacle in principle to its generalization to other rational CFTs, and it then suggests that the 1-point functions of bulk fields between suitable boundary states are determined by purely topological data of the fusion algebra. It also gives a possible way to *define* (at least ratios of) the fusion rules for non-rational CFTs.

# 4 Conclusion

We have proposed using smeared boundary states as trial variational states for massive deformations of CFTs. This is motivated by the uses of these states in quantum quenches and entanglement studies. In the case of the 2d minimal models we can perform explicit calculations which show this method gives a qualitative picture of the phase diagram in the vicinity of the CFT. Its main failing is that it cannot correctly predict a flow to a non-trivial CFT. In this case it appears to suggest phase coexistence rather than a continuous transition. In addition, the boundaries between different states corresponding to different renormalization group sinks are always first-order transitions. This is a necessary consequence of the variational method.

However the method, by its nature, always gives the correct scaling of the energy with the coupling constants. From a numerical point of view it cannot be competitive with earlier methods such as the truncated conformal space approach [18,19], but it is much simpler and moreover gives new insight into the physical relationship between conformal boundary states and ground states of gapped theories. Since it gives a bound on the universal term in the free energy, it would be interesting to make a detailed comparison with exact results available for integrable perturbations [20].

# Acknowledgements

The author thanks V. Pasquier, H. Saleur and G. Vidal for helpful discussions, and A. Konechny for pointing out Ref. [7].

**Funding information:** This work was supported in part by funds from the Simons Foundation, and by the Perimeter Institute for Theoretical Physics. Research at the Perimeter Institute is supported by the Government of Canada through the Department of Innovation, Science and Economic Development and by the Province of Ontario through the Ministry of Research and Innovation.

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
