# Peer review of "Bulk Renormalization Group Flows and Boundary States in Conformal Field Theories"

_SciPost Physics, doi:SciPost Phys. 3, 011 (2017)_

## Round 3 · Referee Report · Anonymous (Referee 1) · 2017-7-13

Strengths

  1. Interesting and novel idea to obtain non-perturbative information about perturbed conformal field theories.
  2. Implementing the idea is simple, and is demonstrated to give relevant and interesting information about dynamics.
  3. Main considerations valid in all space-time dimensions.

Weaknesses

  1. Clarity of some parts of the text needs improvement.
  2. A potential and natural comparison to previous results in the literature is missing.

These points are detailed in the requested changes.

Report

The paper proposes a novel and rather nice idea to obtain the ground states of perturbed conformal field theories, which is valid in any dimensions, albeit it is explicitly demonstrated only in 1+1 dimensional spacetime. It is shown to reproduce known features of simple perturbed minimal models, but the potential applications of the idea are eventually much wider, and give a general and simple first approximation for many other models, including non-integrable ones. As a result, the paper is expected to have a number of applications and therefore significant impact, and definitely deserves to be published in Scipost Physics, after addressing some weaknesses of presentation (cf. requested changes) and considering a potential comparison to previous results in the literature.

Requested changes

Comparison to previous results, suggested for inclusion in the paper:

since the method eventually results in an estimate of the vacuum energy density, it is of interest to compare it to existing results in the literature. A suitable class of examples seems to be the $\Phi_{1,2}$/$\Phi_{2,1}$ perturbations of minimal models, for which the exact ground state energy density was computed in terms of the coupling, cf. the paper

V.A. Fateev: The Exact relations between the coupling constants and the masses of particles for the integrable perturbed conformal field theories, Phys. Lett. B324 (1994) 45-51.

Suggestions to improve the presentation:

  1. In the Introduction (second paragraph, page 3) the author states that in simple cases the relevant conformal boundary condition is apparent by physical inspection. The example brought up is scaling Ising field theory. However, the argument looks unfinished, the conclusion regarding the relevant boundary condition for the different perturbation is not clearly stated.

  2. Still in the Introduction, after eqns. (5) and (6) the argument starts for general dimension D+1, then switches to 1+1 when the author describes his interesting side result concerning matrix elements of primary operators between Ishibashi states. But this switch is not indicated in the text, which may be confusing for the reader.

  3. In Section 2 (General formalism) the operators T, Tbar in eqn. (11) are undefined in terms of the energy-momentum tensor $T_{ij}$ introduced above. While this is standard CFT formalism, a one-line definition would improve the reading of the text.

  4. Similarly, in the argument starting with eqn. (16) a figure specifying the relevant directions of quantization for the reader would help.

  5. After eqn. (26) including explicitly the resulting scaling $E_a \rightarrow e^{(1+D)l} E_a$ would also be helpful for the reader.

And finally a typo: after eqn. (23) the work "set" is missing in the phrase "complete of eigenstates".

  • validity: top
  • significance: high
  • originality: high
  • clarity: good
  • formatting: excellent
  • grammar: perfect

Author:  John Cardy  on 2017-07-14  [id 153]

(in reply to Report 1 on 2017-07-13)
Category:
remark

I thank the author of the report for his/her constructive suggestions on the text. A comparison with the exact results of Fateev is of course possible but it will take some time to ensure that conventions (eg factors of 2\pi) are in agreement.

---

## Round 3 · Referee Report · Anonymous (Referee 2) · 2017-7-21

Strengths

See report

Weaknesses

See report

Report

The paper is devoted to the use of smeared boundary states of conformal field theory for approximating the ground state of massive field theories. The idea may not appear natural in view of the fact that the direct way to describe such ground states in the basis of states of the ultraviolet conformal field theory - the truncated conformal space approach - is well known and very effective. As the author explains, the motivation comes from recent studies such as quantum quenches in conformal field theory. In this particular version of the quench problem one would like to follow the massless time evolution of a closed system initially prepared in the ground state of a massive theory. While the problem remains unsolved, the author showed some time ago that a variant amenable to analytic treatment is that in which the initial state is replaced by a smeared conformal boundary state. The present study then goes in the direction of investigating to which extent the tractable initial state can mimic the physical one.

The tractable initial state is a conformal boundary state evolved with the conformal Hamiltonian for an imaginary time \tau which mimics the length scale of the massive ground state. The author treats these "smeared" boundary states as variational approximations for the massive ground states and performs explicit calculations for the case of minimal models of two-dimensional conformal field theory, for which he obtained in the past a correspondence between conformal boundary states and bulk operators. He finds that a variational ansatz with \tau and the boundary state as variational parameters yields a reasonable qualitative picture of the phase diagram near the conformal point. A main intrinsic limitation appears to be the inability of the method to account for massless flows, a price to pay for analytic tractability. Concerning the quench problem, the relevance of smeared conformal boundary states depends on the sensitivity of the unitary long time evolution to the initial condition, an issue poorly understood at present.

Requested changes

No changes requested

---

## Round 3 · Referee Report · Paul Fendley (Referee 3) · 2017-7-26

Strengths

1- a new approach to variational states in field theory, elegant and simple

2- a possibly deep relation to topological properties of conformal field theory

Weaknesses

1- nothing serious, a few references should be added and a few extra comments made

Report

Inspired by previous work on quantum quenches and the entanglement spectrum, Cardy uses conformal field theory to make a variational ansatz for the ground state of a massive field theory. The massive field theory must be defined as a perturbation of said conformal field theory. One key observation is that the boundary states in conformal field have very special properties, for example yielding nice (and computable) one-point functions. As he points out, boundary states are not suitable for variational states of a massive theory, but need to be "smeared". He does this (where the inspiration from quenches enters) by acting with the Hamiltonian. In the set-up of an RCFT, everything at least in principle then can be computed. This is done for the minimal models here, and the physical consequences are nicely in accord with known intution.

A very interesting consequence is that any particular perturbation corresponds to a particular boundary state, so that the full RG trajectory is along a ray in this space. This is the standard intution, but it is beautiful to see it work out explicitly in this ansatz.

In general, the strength of this approach is that it is quite intuitive and quite simple to understand. I thus think this paper should be published. Even though, as noted by the author, the ansatz as it stands does yet allow for flows to massless states. Since these are notoriously difficult by any approach, so this isn't a big flaw. It does present an interesting future project; obviously these RG trajectories are very special, and so one would hope that the corresponding states have some identifiable special properties.

Requested changes

All are minor:

1- As I said above, this should work for all RCFTs. In fact, the somewhat mysterious but very interesting equation 55 makes it clear that all the data involved is ultimately topological. Nevertheless, for reasons I don't quite follow Cardy restricts to minimal models. If this is simply for technical ease, that's fine, but the way I read the text is that there seems to be some obstacle to making it general. Thus if there is an obstacle, this should be commented on, or if not, it should be clarified.

2- A few more comments/speculations on potential applications would be nice. Could the ansatz be tested by say computing one-point functions using it and comparing with exact results in the integrable cases?

3- The analysis of the Phi_{1,3} perturbation of the minimal models provides a major check on these results, in that it reproduces the correct (high) ground-state degeneracy. This degeneracy was described long ago by Huse in Phys.Rev. B30 (1984) 3908-3915 . This should be highlighted!

4- It is stated that having positive Casimir energy is physically reasonable, and indeed that is true. However, in certain geometries the Casmir effect is repulsive (see Levin et al arXiv:1003.3487 ). Is there any relevance of that story to these states (in higher dimension than 1+1 presumably)?

5- Affleck wrote a paper cond-mat/0005286 on boundary states and flows between them in the tricritical Ising model in terms of the much more intuitive Blume-Capel model instead of the integrable A_4 one. This might be useful in lending some intution to fig. 2.

6- Sorry, I didn't find fig. 1 comprehensible. A few more words, maybe? In fact, while there is a fairly brief comment on the similarity to matrix product states, this might be worth a little more exploration. Indeed this approach is not likely to be as accurate numerically as DMRG etc, but given the much stronger intution here, making a connection might be quite fruitful.

7- A few more pointed references to the old literature or to review articles on the old boundary CFT story would be useful (e.g. to the author's 1989 paper, to Moore and Seiberg on the topological bits of CFT, various Les Houches lectures, etc).

8- In the "Casimir" paragraph finishing top of p. 7, it should note that "excited state" is the lowest-energy state in the sector with (ab) boundary conditions (i.e. restate the last sentence in the CFT language)

---

## Round 4 · List of Changes

In response to Anonymous Report 1:

- Comparison with work of Fateev: sentence added at end that this would be interesting to do.
- on p.3, discussion of correspondence between RG sinks and boundary states in Ising expanded.
- after eqs (5,6) it is made clear when I am specializing to 1+1 dimensions.
- in sec.2, T and Tbar are defined.
- after eq, 6 it is explained in the text what is the direction of quantization, rather than introducing a new figure.
- in eq (26) the rescaling of E_a is stated explicitly
- typo after eq (23) corrected.

In response to Paul Fendley:

1. It should work in principle for all RCFTS but e.g even the boundary states haven't been worked out in general. I have inserted wording that there is no obstacle in principle, as far as I know.

2. Are 1-point functions known for boundary states in integrable cases? I don't think so.

3. Thanks, this point now emphasized and reference to Huse added at this point.

4. Levin et al state in their first sentence: `the Casimir force between parallel plates is attractive'. They then look at other geometries, not relevant to this work.

5. Thanks, I have now included discussion using Affleck's identification of boundary states, which I agree is more intuitive.

6. I have added to the caption, hopefully making what is a rather stretched comparison (which came from a comment from G Vidal) more comprehensible.

7. I added references to my 1989 paper and also a good review by Petkova and Zuber.

8. Thanks, yes, corrected.

---

## Editorial Decision

published